# Clinical Findings, Management, Imaging, and Outcomes in Sea Turtles with Traumatic Head Injuries: A Retrospective Study of 29 *Caretta caretta*

**DOI:** 10.3390/ani13010152

**Published:** 2022-12-30

**Authors:** Delia Franchini, Serena Paci, Stefano Ciccarelli, Carmela Valastro, Pasquale Salvemini, Antonio Di Bello

**Affiliations:** 1Department of Veterinary Medicine, University of Bari “Aldo Moro”, SP 62 per Casamassima Km 3, 70010 Valenzano, Italy; 2WWF Molfetta Rescue Center, Via Puccini 16, 70056 Molfetta, Italy

**Keywords:** sea turtle, *Caretta caretta*, head trauma, neurological scores, CT

## Abstract

**Simple Summary:**

Sea turtles are considered an endangered species, largely due to anthropogenic activities. Severe trauma in these species mainly involves the carapace and head. Studies on the incidence of head trauma in sea turtles and analysis of the survival rate after hospitalization performed on many subjects examined over a long period of time were missing in the scientific literature. In this retrospective study, we evaluated 1877 *Caretta caretta,* 29 of which showed head trauma. The severity of head injuries was assessed clinically and by neurological examination. CT examination was essential to evaluate the impaired central nervous system and sense organs. According to our scoring classification, most head traumatized turtles had severe injuries but in most cases, the animals showed no alteration in mentation state. Only 28% (8/29) of the turtles showed head damage related to severe neurological deficits. Indeed, 21 out of 29 sea turtles were released after a time ranging from a few days to 8 months. To the best of our knowledge, the literature lacks specific data on the incidence, correlations with neurological deficits, complications, and survival rate of traumatic head injuries in loggerhead sea turtles.

**Abstract:**

Sea turtles are considered endangered species, largely due to anthropogenic activities. Much of the trauma in these species involves the carapace and skull, resulting in several degrees of damage to the pulmonary and nervous systems. Among traumatic injuries, those involving the skull can be complicated by brain exposure, and turtles with severe skull injuries that have nervous system impairment, emaciation, and dehydration can often die. Between July 2014 and February 2022, a total of 1877 loggerhead sea turtles (*Caretta caretta*) were referred for clinical evaluation at the Sea Turtle Clinic (STC) of the Department of Veterinary Medicine of the University of Bari. A retrospective study of 29 consecutive cases of loggerhead sea turtles (*Caretta caretta*) with skull lesions of different degrees of severity is reported. On admission, physical and neurological evaluations were performed to assess and grade the lesions and neurological deficits. In 20 of the 29 sea turtles with more serious head trauma, computed tomography (CT) findings in combination with physical and neurological assessment enabled the evaluation of the potential correlation between deficits and the extent of head injuries. All sea turtles underwent curettage of the skull wounds, and the treatment protocol included the use of the plant-derived dressing 1 Primary Wound Dressing^®^ (Phytoceutical AG, Endospin Italia) applied on the wound surface as a primary dressing. Out of 29 sea turtles, 21 were released after a time ranging from a few days to 8 months. To the best of our knowledge, the literature lacks specific data on the incidence, correlations with neurological deficits, complications, and survival rate of loggerhead sea turtles with traumatic head injuries.

## 1. Introduction

Nearly all species of sea turtles are classified as vulnerable because of their high degree of exposure to anthropogenic effects [1]. Traumatic injury to sea turtles can occur from several causes such as vessel strikes, interaction with dredges, fishing activity, fishermen, water-controlling equipment, sharks, and other turtles [2]. Damage to the lungs or brain by penetration of fracture bone fragments is a common consequence of carapace or skull fracture [3,4,5,6,7]. The relatively small volume of the braincase compared to the size of the head and protection by the large muscles of mastication and the outer skull results in wounds that may not immediately impair vital function, but result in permanent impairment of vision, vestibular function, and other senses [8]. Meanwhile, blunt trauma to the head can result in cerebral edema or intracranial hemorrhage, with or without evident fracture and deep soft tissue lacerations [9]. If the sea turtle survives, weakness, disorientation, and irreversible deficits may ensue, hindering the turtle’s ability to feed or escape from predators [10]. The sea turtles presented for clinical evaluation for severe head trauma are emaciated and anorexic. Clinical reports describing the treatment of head injuries resulting from trauma in sea turtles are rare [4,5,6,7]. Surgical curettage of traumatic head wounds and subsequent daily use of topical dressing was found to be favorable in reducing opportunistic infections, promoting soft tissue healing, and improving the clinical conditions of these animals and their survival rate [4]. Because of the severe debilitation of sea turtles with severe head trauma, mortality may occur during rehabilitation. The prognosis is poor if the turtle is incredibly depressed and unable to be supported in the water and feed appropriately [11]. An early diagnosis and correct staging of traumatic head injuries of sea turtles by clinical examination and, in severe cases, CT of the skull are useful for classifying the extent of the wounds and the possible involvement of the brain tissue. To the best of our knowledge, preexisting literature lacks specific data on the incidence, correlations with neurological deficits, complications, and survival rate of loggerhead sea turtle’s traumatic head injuries. In July 2014 we began collecting all cases of head injuries in sea turtles referred to the STC of the Department of Veterinary Medicine of the University of Bari, with the purpose of studying: (1) the incidence of head trauma in sea turtles, (2) the severity and type of injuries and the role of CT in their evaluation, (3) the neurological deficits and damage to the sense organs and whether there is a correlation between these deficits and the extent of head injuries, (4) the treatment, (5) the sequelae and complications of this condition, and (6) the rate of mortality.

## 2. Materials and Methods

Between July 2014 and February 2022, a total of 1877 loggerhead sea turtles (*Caretta caretta*) were referred for clinical evaluation at the STC, transferred from Adriatic and Ionian Sea turtle rescue centers. The data of all consecutive cases of head injuries were analyzed. We recorded whether turtles with head injuries were found adrift, stranded, or by-caught. Upon admission, a physical examination of each turtle was performed; curve carapace length (CCL), curved carapace width (CCW), weight, Body Condition Score (BCS) evaluation, measurement of core body temperature from the cloaca, and blood sampling to measure hematology and biochemistry profile. Head injuries were assessed and classified as acute or chronic (Figure 1). A score from 1 to 3 was assigned: 1. mild (small extension, superficial wound, linear skull fracture or mild depression of bones, absence of underlying soft tissue exposure); 2. moderate (medium extension, non-linear bone fracture with the presence of bone fragments that do not penetrate the underlying soft tissues despite minimal exposure); 3. severe (severe skull fracture with penetrating bone fragments and possible brain exposure and bleeding).

A neurologic examination was performed to evaluate neurological deficits. General activity, neck, front and rear flipper movements, and pupillary, eyelid, menace responses and cloaca reflexes were observed. Mentation was assessed and classified in three degrees: 1. alert (responsive to external stimuli); 2. depressed (reduced, but appropriate responsiveness to external stimuli); 3. lethargic/comatose (not responsive, loss of consciousness) [12]. Loggerhead sea turtles underwent a total body radiograph in dorso-ventral (DV), latero-lateral (LL) for each side, and craniocaudal (CrCd) radiographic examinations plus dorso-ventral (DV) and latero-lateral (LL) of the skull. In cases of moderate/severe head injuries and all cases of neurological symptoms or involvement of the sense organs, a multi-detector computed tomography (MDCT) with a 16-slice MDCT scanner (Somaton Emotion, Siemens, Forchheim, Germany) was performed. The technical scan and reconstruction parameters were 110 KVp, 180 mAs, 1 mm slice thickness, pitch of 0.8, 0.6 s/rotation, 0.5 mm reconstruction interval, and standard (bone) acquisition algorithm. Three-dimensional (3D) multi-planar reformatted, maximum-intensity projection and volume-rendered images were obtained using a dedicated 3D software program (Pixmeo OsiriX DICOM viewer^®^, Pixmeo, Bernex, Switzerland) (Figure 2).

The turtles underwent anesthesia for debridement and curettage of the skull wounds, and in cases where turtles were unable to eat by themselves, an esophagostomy tube was surgically inserted [13]. Swabs for microbiological examination were collected from head wounds to identify opportunistic infections and when necessary, set up targeted antibiotic therapy. Following surgical curettage, the treatment protocol included the rinsing of the wound with sterile saline and the exclusive use of the plant-derived dressing 1 Primary Wound Dressing^®^ applied on the wound surface as a primary dressing, daily for the first month and then every other day until the end of treatment [4]. In severely traumatized turtles with concomitant pathologies (entanglement, hooks, and lines in the gastrointestinal tract, pulmonary diseases) or very weak, antibiotic therapy with enrofloxacin (Baytril 5%, 5 mg kg−1 IM after dilution 1:4 in NaCl solution 0.9%, every 24 h) was initiated and possibly modified after the outcome of the antibiogram. Tramadol (Altadol, Formevet^®^, Milano, Italy; 5 mg/kg) [14] was administered intravenously in the cervical venous sinus as a pre-anesthetic, and after 20 min, anesthesia was induced with 3–7.0 mg/kg intravenous propofol (Propovet^®^, Zoetis, Rome, Italy). A 4- or 7-mm diameter endotracheal tube was used for tracheal intubation, and anesthesia was maintained, when necessary, with oxygen and sevoflurane 2 to 3% (SevoFlo, Ecuphar^®^, Greifswald, Italy) throughout the surgery. Anesthesia was monitored by electrocardiography and capnography. A catheter was placed in the cephalic vein [15], and lactated Ringer’s solution and 0.9% NaCl (1:1) were administered at 2 mL kg/h. Pain management was achieved by administration of tramadol (Altadol, 5 mg/kg IM, every 48 h) for 3–7 d, after surgical debridement. Dexamethasone (Dexadreson 0.2 mg/kg IM, every 48 h) therapy has been used in subjects with severe head trauma and cerebral edema detected by CT [16,17]. During the 24 h following surgery, each turtle was kept wrapped in damp cloths in a few centimeters of warm water. By the second day, the sea turtles were kept in tanks with saltwater at 36–37‰ (salinity percentage of the Adriatic and the Ionian Sea, respectively) in a room at approximately 23–25 °C until clinical improvement. Sea turtles with mild neurological signs and showing the ability to pull their heads out of the water were placed in deep water; turtles with moderate or severe neurological signs showing difficulty in breathing were placed in tanks with foam mattresses, with saline water that did not exceed the nostrils.

## 3. Results

Out of the 1877 sea turtles that were referred to STC for a clinical evaluation over the course of seven years, 29 had head trauma. The curve carapace length (CCL) from notch to tip ranged from 22 to 75.5 cm (mean 61.5 cm), curved carapace width (CCW) from 20 to 70 cm (mean 56.4 cm), and weight from 1.1 to 50 kg (mean 25.9 kg). Considering the size and expression of sexual dimorphism, 6 out of the 29 sea turtles with head trauma were juveniles, 14 were subadults, and 9 were adults. On a scale from 1–5, the BCS was normal (3) in 10 animals, while 14 were thin (2) and 5 were emaciated (1). Measurement of core body temperature from the cloaca ranged from 18 °C to 22°C. Out of these 29 sea turtles, 21 had been found stranded along the coasts, while 6 floated adrift and 2 were by-caught. At the time of presentation to the STC, 7 out of 29 traumatized sea turtles showed clinical signs of acute trauma, such as active bleeding, oozing of serum, sharp wound margins, and lack of necrosis while 22 showed wounds with signs of chronicity with the presence of fibrinonecrotic exudate, necrotic tissue, and blunt wound margins. External head injuries were classified, and 4 turtles presented with mild, 6 with moderate and 19 with severe trauma. Mentation was classified as alert in 14 turtles, depressed in 9, and lethargic/comatose in 6 (Figure 3). In 10 subjects, the head trauma involved the orbital, nasal, and squamosal bones with a possible compromise of the sensory organ (eyeball, nose, ear) (Figure 4).

Out of the 29 traumatized turtles evaluated, 2 had concomitant entanglement of the front flipper and neck, 2/29 had ingested hooks and lines, and 4/29 had pulmonary diseases (Figure 5).

During the evaluation into the water, 6 turtles showed symmetrical, or asymmetrical (upward left or right side) buoyancy abnormalities, and 18 turtles showed an inability to submerge (Figure 6).

According to the radiographic examination, all sea turtles presented open and comminuted fractures, and depressed fractures with or without loss of skull tissue.

Total head and body examinations by CT scan were performed on 20 turtles. Sea turtles with a neurological score of 3 (*n* = 6) did not undergo general anesthesia for CT scan examination, those with a neurological score of 1 *(n* = 14) received general anesthesia, with 5 mg/kg propofol (Propovet^®^, Zoetis, Rome, Italy) injected intravenously (IV) via the external jugular vein. The images were then submitted to 3D and MPR (OsiriX, Pixmeo Sàrl). In severe cases, evaluation of the CT images revealed the presence of depressed comminuted fractures of the parietal, postorbital, prefrontal, frontal, orbital, nasal, and squamosal bones. All lesions were most lytic with no formation of new bone tissue. In eight turtles, some of the fragments appeared inverted toward the disrupted dura, and pneumocephalus and cerebral lacerations and cerebral swelling were detected (Figure 7).

In six cases, the ocular globe was shifted downward due to the pressure of the surrounding tissue resulting in brain contusion and compression of the salt gland. In moderate and mild cases, parietal and postorbital fragments were deeply or superficially depressed, respectively (Figure 8).

In some cases, portions of the adductor muscles of the jaw (pseudotemporal and dorsal and ventral pterygoid) were lacerated.

Culture and susceptibility tests were performed on wound lesions of all 29 loggerhead sea turtles. Gram-negative microorganisms were found and identified as follows: *Pseudomonas paucimobilis, Pseudomonas putrefaciens, Pseudomonas stutzeri, Vibrio fluvialis, Citrobacter freundii, Acinetobacter calcoaceticus, Enterobacter* sp., *Proteus* sp. Sea turtles underwent anesthesia for debridement and curettage of the skull wounds, and an esophagostomy tube was placed in 17 sea turtles on the 3rd day because they were unable to eat by themselves [13]. Following surgical curettage, 1 Primary Wound Dressing^®^ was applied on the wound surface in a thin layer directly on the wound as a spray [4]. A specific antibiotic therapy was administered in 19 severely traumatized sea turtles for 15 days, respectively, based on enrofloxacin (5 mg/kg, IM, SID) in 18, and Ceftazidime (22 mg/kg, IM, EOD) [18,19] in two turtles. Dexamethasone (0.2 mg/kg IM SID) was administered to four turtles with severe trauma and brain edema highlighted at CT. In 24 sea turtles with acute trauma or moderate/severe injuries, tramadol (5 mg/kg, IM, SID) was used as analgesic support before and after curettage. Out of 29 turtles, during the rehabilitation period, 8 (*n* = 5 after a few days and *n* = 3 within a month from hospitalization) exhibited severely altered neurological status and died. Out of 29 sea turtles, 21 were released after a time ranging from a few days, in cases of grade 1 head injuries, to a range of 1-8 months, in grade 2 and 3 cases (Figure 9) (Table 1).

## 4. Discussion

Accidents involving propeller or hull impact are a common cause of injury and death among sea turtles. Basking juveniles are also predisposed to boat strikes, as they feed in shallow water relatively close to the shore [10]. Moreover, fishermen may deliberately traumatize sea turtles presumed to have decreased catches or damaged gear [16]. In this study, we considered significantly large number of sea turtles over 7 years with 1.5% (29/1877) of animals affected by head trauma in the Adriatic and Ionian waters of the Mediterranean Sea. Early diagnosis and treatment of sea turtles with head trauma improves the morbidity and mortality rates [20], but unfortunately, most of them are not evaluated immediately after trauma. In our study, only a quarter of sea turtles had acute lesions at the time of evaluation and in good nutritional status, while the others showing chronic lesions appeared thin or emaciated. Our data showed that two-thirds of sea turtles with head injuries were found stranded along the coasts or floating adrift and were almost always very weak animals unable to dive normally and feed. If not rescued these animals are doomed to certain death.

The total body radiographic examination is useful and should always be performed to highlight concomitant pathologies (e.g., hooks and lines in the gastrointestinal tract, pulmonary diseases), but skull X-ray is not indicated for the study of cranial trauma due to the overlapping of the various bony structures of the head and the impossibility to assess soft tissue such as the brain. For this reason, a CT scan examination should be performed in animals with severe head injuries, especially in depressed fractures of the cranial vault, extensive loss of bone, deep fractures, or in turtles with neurological signs and the involvement of the sense organs.

The skeletal structures frequently affected by the trauma were in mild cases the parietal and frontal bones, while in moderate cases were also the postorbital bones and in severe cases, even the nasal, maxillary, premaxillary, jugal bones, nasal, and cranial cavities were affected.

According to our scoring classification set for the grading of skull injuries, 65% (19/29) were severe lesions, while 21% (6/29) and 14% (4/29) were moderate and mild wounds, respectively. In turtles with severe head trauma, skull fragments can cause impact injury to underlying brain tissue.

Severe head trauma may cause brain exposure and damage salt glands or involve the sense organs [6,21]. Traumatic injuries involving the postorbital regions could damage the underlying salt glands, involving their essential function in electrolyte homeostasis [22]. The relatively small volume of the braincase compared to the size of the head and protection by the large muscles of mastication and outer skull, results in wounds that may not immediately impair vital function, but that may impair permanently vision, vestibular function, and other senses. Sea turtles do not have external orifices associated with the ears, but the middle ear is located within the temporal region, covered by a simple tympanum formed of scales overlying fibrous connective tissue. For this reason, any trauma affecting the lateral region of the skull could compromise hearing and vestibular perception in sea turtles. According to the authors’ clinical experience, sea turtles hospitalized for entanglement injuries to the front flippers have a greater chance of concomitantly presenting lateral skull injuries and ears damage, due to repeated trauma from the litter that often gets trapped in the line or gear which has caused entanglement.

Due to the anatomy of sea turtles, it is difficult to separate taste and olfaction in these animals. Any disorder that damages or hinders the sensory fibers in the nasal or oral epithelium could potentially affect the animal’s willingness to find food or its desire to eat. In our study, 35% (10/29) of the traumatized sea turtles displayed signs of sense organ impairment, including blindness, which made it harder for them to eat. Therefore, observing responses to offered food items is a good general means of assessment of sensory ability and response [8]. Since many turtles undergoing rehabilitation were thin to emaciated, gaining weight as muscle and fat stores were the top priority [23]. Based on physical examination, body condition and blood parameters, tube feeding was required in 57% of anorexic turtles.

A complete neurological examination, including observations made both in and out of the water, performed in all 29 traumatized sea turtles revealed that 48% (14/29) had no impairment of the nervous system (NS) and mentation state was alert; in 31% (9/29), the mentation state was depressed, and in 21% (6/29) the mentation state was lethargic/comatose. The comparison between the type and severity of injuries and the neurological evaluation of these animals showed there is not always a correlation between the severity of the head injury and the development of serious neurological deficits or alteration of the mentation state. In the present study, 28% (8/29) of the turtles showed head damage related to severe neurological deficits. A previous study has documented the relationship between neurological status, trauma to the carapace, and CT findings in 10 sea turtles of multiple species [3]. The neurologic deficit was monitored during the hospitalization period. In animals that have survived, we observed a progressive recovery of neurological reflexes during or after complete healing of the head.

Bacterial infections caused by Gram-negative opportunistic pathogens are commonly reported in free-ranging and captive sea turtles, but many of these microorganisms could be dangerous in traumatized sea turtles [24]. We performed microbiological tests after surgical curettage of head injuries. Fifteen bacterial isolates were made that were subsequently subjected to evaluation of their antibiotic resistance profile.

Resistance appeared common to b-lactam antibiotics (18/20) and tetracyclines (17/20).

For severe lesions with exposure of the brain tissue, euthanasia often appears to be the most ethical option [10]. Performing a surgical curettage under general anesthesia with the subsequent application, repeated during the dressings of a plant-derived dressing 1 Primary Wound Dressing^®^ has been shown to have a marked capacity to promote re-epithelialization and wound healing in mammals, even in cases that are very difficult to manage [4,25,26]. In most cases, head trauma caused turtles to die immediately or during rehabilitation [22].

Sensory function is a significant consideration in the decision of whether a sea turtle is a viable candidate for reintroduction into the sea. Our pre-release evaluation included an examination of whether these animals were capable of locating and swallowing food. Loss of vision in one eye may diminish general fitness, but blind turtles are impaired in their ability to survive. Similarly, permanent vestibular dysfunction that interferes with swimming and buoyancy is a non-releasable condition. In our study, the buoyancy disorders resolved during the hospitalization, and the lesions affecting the function of the sense organs were not considered so disabling for the animal as to prevent its release into the wild. The mortality rate was high with 28% (8/29) of sea turtles that had arrived at STC and died during the rehabilitation period, as they were exhibiting severely altered neurological status and brain tissue exposure.

## 5. Conclusions

Many sea turtles die in the sea due to severe brain damage or starvation as a consequence of an inability to feed following injuries disabling the sense organs and swimming ability. The literature lacks specific data on the incidence, correlations with neurological deficits, complications, and survival rate of traumatic head injuries in loggerhead sea turtles. In this retrospective study, we evaluated 29 *Caretta caretta* with head trauma. The severity of head injuries was assessed clinically and by neurological examination, and CT examination was essential to evaluate the impaired central nervous system and sense organs. Most head-traumatized turtles had severe injuries, but in the majority of cases, the animals showed no alteration in mentation and only 28% showed severe neurological deficits. After management of the head injuries and rehabilitation period, 72% of sea turtles were released. Since no data has been collected following the release of turtles who have been treated for skull trauma, in the future satellite tracking of these animals would be an aid to evaluate the continuing evolution of treated head injuries and to estimate the long-term survival rates of these turtles.

Hundreds of traumatized turtles are rescued and treated annually in rescue centers.

In a stranded or drifting sea turtle with a head injury, it is essential to perform an early and accurate diagnosis, with consequent appropriate treatment improving prognosis, minimizing brain injuries, and increasing survival.

## Figures and Tables

**Figure 1 animals-13-00152-f001:**
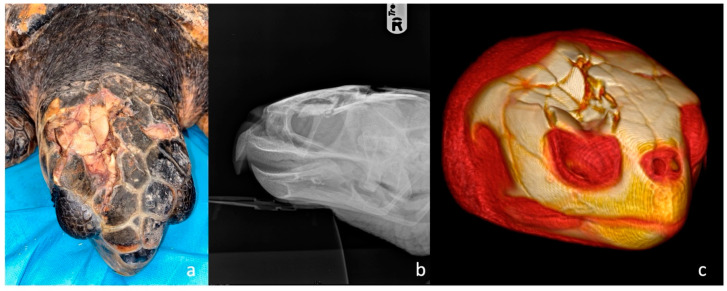
Head trauma staging example (**a**) Clinical assessment of a grade 3 skull trauma; (**b**) X-ray in latero-lateral view showing parietal and postorbital fractures; (**c**) 3D-CT scan of the same turtle’s skull showing the same clinical pattern and the compromised right ocular globe.

**Figure 2 animals-13-00152-f002:**
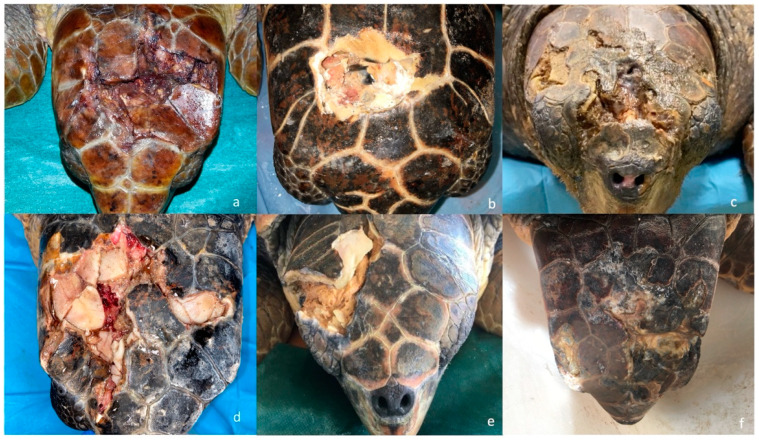
*Examples of Caretta caretta with head trauma arrived at STC*. (**a**) Acute trauma of parietal and frontal bone with hemorrhage; (**b**) chronic trauma of parietal bone with exposure of the underlying soft tissues; (**c**) chronic trauma of frontal, nasal bone and right postorbital bones; (**d**) acute trauma of right left parietal, frontal, postorbital bones with oozing of serum and hemorrhage; (**e**) chronic trauma of right postorbital and parietal bones with the presence of fibrotic tissue; (**f**) chronic trauma of left frontal bone and parietal bone.

**Figure 3 animals-13-00152-f003:**
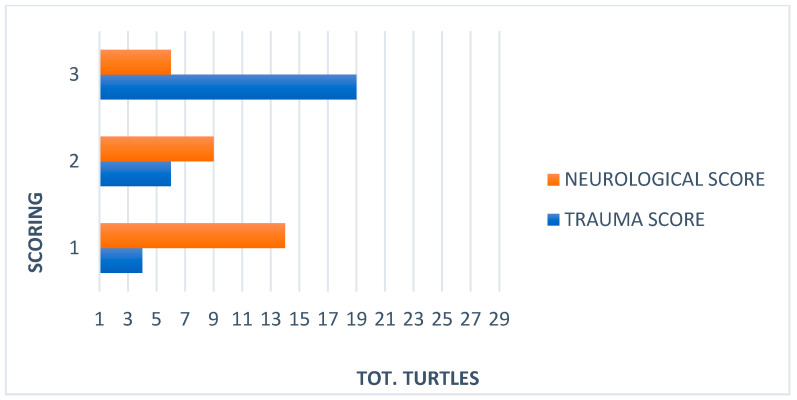
Comparison between trauma and neurological scores in 29 *Caretta caretta*.

**Figure 4 animals-13-00152-f004:**
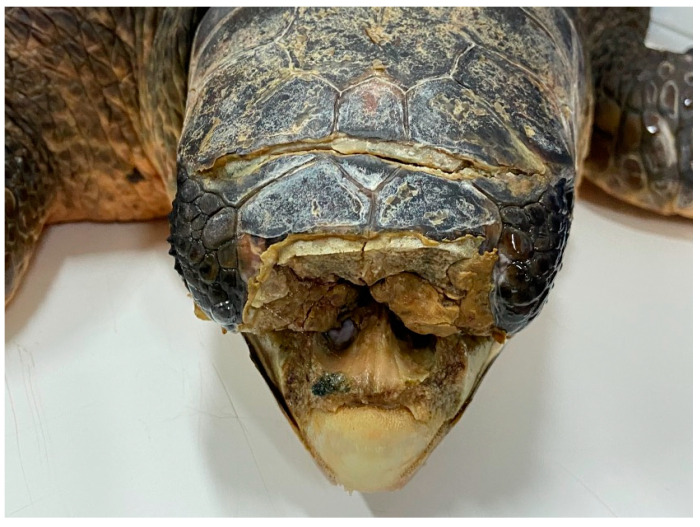
*Caretta caretta* with fracture of the frontal and nasal bones and massive loss of bone tissue with impaired sense organs and exposure of the nasal cavities.

**Figure 5 animals-13-00152-f005:**
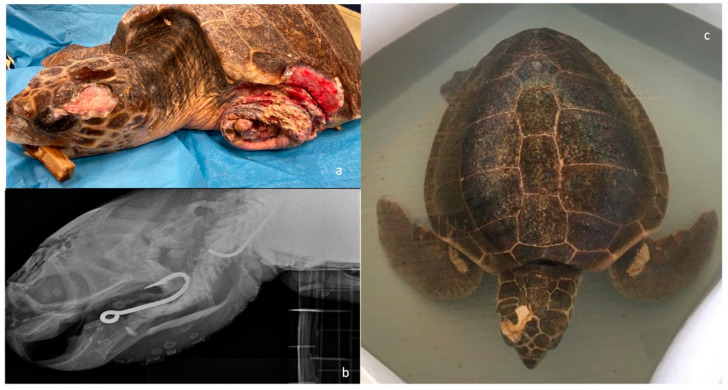
Examples of concomitant pathologies in sea turtles hospitalized for head trauma at the STC. (**a**) Severe entanglement of the left anterior fin resulting in amputation. (**b**) X-Ray in latero-lateral view of the skull of a sea turtle with skull fracture and presence of a large hook inside the buccal cavity and one at the level of the cervical esophagus. (**c**) Sea turtle with head trauma, buoyancy disorders, and entanglement of both front flippers.

**Figure 6 animals-13-00152-f006:**
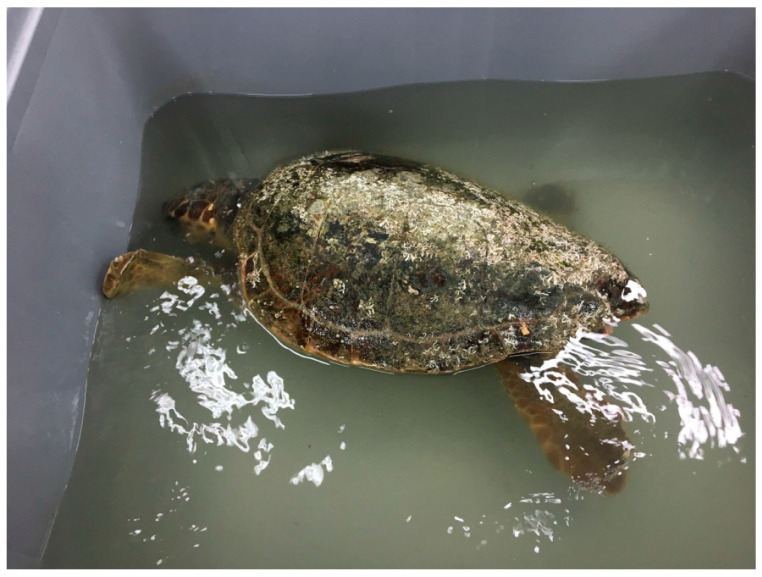
*Caretta caretta* with buoyancy disorders and difficulty diving.

**Figure 7 animals-13-00152-f007:**
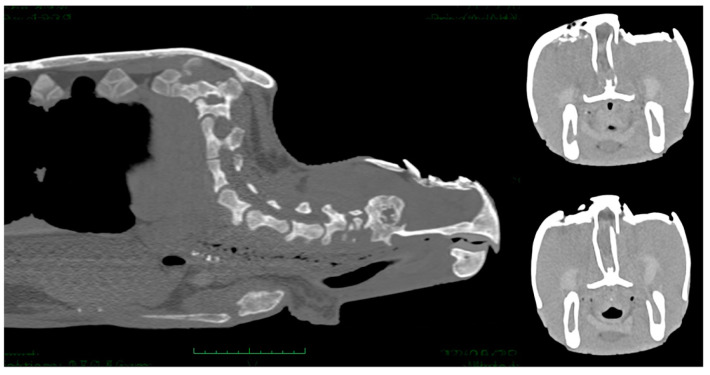
CT scan images showing comminuted fractures of the parietal, postorbital, prefrontal, and frontal bones. The lesions were lytic and skull fragments appeared inverted toward the disrupted dura, pneumocephalus and cerebral lacerations and cerebral swelling are evident.

**Figure 8 animals-13-00152-f008:**
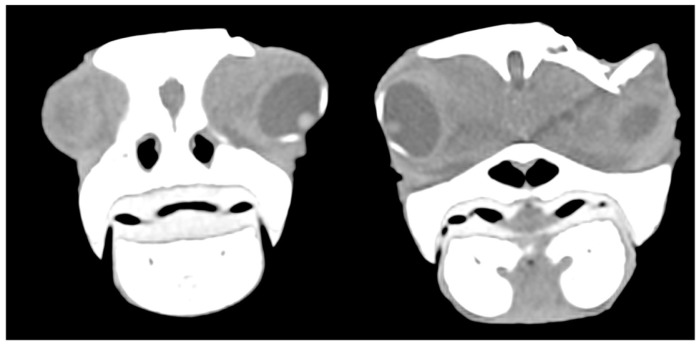
CT scan showing the displacement of the eyeball downwards due to the pressure of the surrounding tissue and fractured postorbital bone.

**Figure 9 animals-13-00152-f009:**
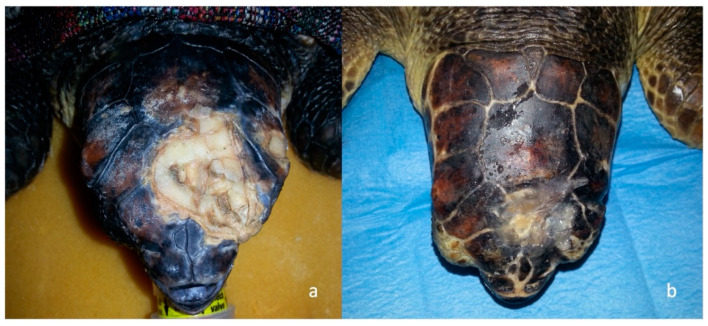
Example of a sea turtle with head trauma treated with surgical curettage and consequent application of 1 Primary wound Dressing^®^. (**a**) Chronic trauma of left parietal and frontal bones with the evident presence of fibrotic tissue; (**b**) The same turtle after surgical curettage and 52 days application of One Vet^®^.

**Table 1 animals-13-00152-t001:** Summary of the result of 29 sea turtles with head injuries: BCS, finding and lesion type, trauma and neurological scores, presence of buoyancy, use of tube-feeding to support nutrition, antibiotic therapy, and outcome.

VARIABLE	BENCHMARKS	N. TURTLES/TOT (29)
BCS	1\5	5
	2\5	14
	3\5	10
	4\5	0
	5\5	0
TYPE OF FINDING	STRANDED	21
	ADRIFT	6
	BY CAUGH	2
TYPE OF LESION	ACUTE	7
	CHRONIC	22
TRAUMA SCORE	1	4
	2	6
	3	19
NEUROLOGICAL SCORE	1	14
	2	9
	3	6
BUOYANCY		6
TUBE FEEDING		17
ANTIBIOTIC THERAPY	ENROFLOXACIN	18
	CEFTAZIDIME	2
OUTCOME	DEAD	8
	ALIVE	21

## Data Availability

Not applicable.

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
