# Peer review of "Clinical Findings, Management, Imaging, and Outcomes in Sea Turtles with Traumatic Head Injuries: A Retrospective Study of 29 Caretta caretta"

_animals, 2022, doi:10.3390/ani13010152_

Round 1
Reviewer 1 Report
Overall the paper is well-written and clear in both its purpose and its results. There are a number of smaller grammatical issues, and these have been included as part of an annotated PDF. Hopefully those are clear, but in their inclusion will help with clarity. Some of the highlighted points are that spacing is not consistent between text and in-text citations, so this should be fixed, presumably with spaces added where needed; anything abbreviated must be clarified with its first use (e.g., BCS); all scientific names need to be italicized, which happens a couple times in text, including with Caretta caretta and with the various microorganisms, along with some occurrences in the figure captions and the references section; I would expect ranges to be listed from low to high (see lines 170-171); I would expect numbers to be listed as n=#, rather than n.#, but that may be a subjective decision; figure captions must be consistent with Figure #. and with a period at the end of the figure caption; make sure to fix line 377, where it presumably should be out of 29 individuals rather than 28; the References section is not consistent in style or format, and this must be reviewed, which I only made note of need for italicizing scientific names and changing two references from having the article titles in ALL CAPS. Finally, the conclusions could be fleshed out a little to act as a partial summary, but the inclusion of the simple summary at the beginning of the paper may serve the purpose enough as it is. One potential addition I think that would help would be to include some graphs or simple representations of the results in a figure. While listing the results is simple enough, representing them graphically would help the reader comprehend the results more quickly and in a more inviting way.

Reviewer 2 Report
Thank you for this study evaluating prognosis of sea turtles with head trauma. Presentation, clinical findings, management, complications and mortality rate are reported. I am not comfortable with your title which doesn't really reflect what is described and reported in your manuscript, to my opinion. I suggest to revise it so that it is more consistent with your data. I also recommend to get some help from a professional English editing service as there is a lot of grammatical and terminology errors. As a general comment, I suggest emphasizing on the various outcomes observed, the large range of lesions noticed and on your management. I had a hard time identifying a take home message at the end of reading the manuscript because there is a lot of information but very little that seems to have a clear impact? Although the information and results are interesting and should be reported, a considerable effort should be made to highlight the real findings of this retrospective study, its interest and what should be learned from it. Also, n my opinion, not all figures are helpful. Here are some more pointed comments on the manuscript itself. However, these only make sense if the general comments I just suggested are taken into account, in my humble opinion.
Abstract:
L12: Please rephrase
L19: I don't think "sensory state" is a correct term. Please prefer "sensorium" or cognitive state; This should be applied to the entire manuscript. Moreover, this sentence is not clear and should be rephrased. Sensorium may be defined as the cognitive or mental state of a patient. Assessment of the sensorium implies interpretation of the patient's consciousness and response to stimuli in the environment. Although an altered sensorium may be appreciated by the clinician. Abnormalities influencing the sensorium can be thought of as affecting the level and/or quality of the mental state. Abnormalities in the level of mentation include depression, obtundation (state of decreased arousal with response to voice or touch), stupor (arousable to vigorous stimuli, but response is incomplete or inadequate), and coma (sustained unresponsiveness to stimuli). Abnormalities in quality include aggression, hyperactivity, hysteria, propulsive movement (the animal may pace or circle), and loss of housebreaking. In these examples, the patient may be fully conscious but may exhibit a change in behavior.
L20: this sentence is not clear either and should be rephrased. There is a mismatch of information.
Abstract
L40: from a few days to 8 months
L50: of fracture bone fragments
L54: prefer blunt trauma
L58: "in most cases", delete this sentence
L63: healing
L65: please rephrase
L75: type of injury
Materials and methods
L80: Caretta caretta
L84: physical examination
L86: hematology and biochemistry profile
L87: delete "accurately"
L89: superficial
L92: delete "their"
L102: delete "compared with normal" as it is obvious
L109: in severely traumatized turtles with concomitant
Results
L168: please rephrase this sentence
L170: 22 to 75.5 cm
L170: 20 to 70 cm
L171: 1.1 to 50 kg
L179: wounds with signs of chronicity
L202: Caretta caretta
L206: Out of the 29 traumatized turtles
L242: "based on the grade of the trauma", what do you mean? what does it have to do with the radiographic evaluation?
L243-244: please reformulate
L246: (n=6)
L247: (n=14)
L252: for which cases? all? it is not clear
L253: this is something you suggest, so this should not be reported in the results section but in the discussion
L299: I suggest "In some cases, portions of the adductor muscles of the jaw (pseudotemporal, dorsal and ventral pterygoid) were lacerated.
L302-304: in how many cases? perhaps add this to the table, and mention the most common organism. Referring to the table will help to find out
L306: was placed
L315: n=5; n=3
Discussion
L342: evaluated
L356: but that may impair permanently vision,...
L365: to separate
L367: willingness?
L369-370: please reformulate
L372-373: please reformulate
L376: delete "of sea turtle"
L376: "sensory state" please refer to comment L19
L381: "sensory state" please refer to comment L19
L398: were capable
Round 2
Reviewer 2 Report
Thank you for having considered my suggestions